# OpenReview forum: "TRAP: Hijacking VLA CoT-Reasoning via Adversarial Patches"
_ICML.cc/2026/Conference — ICML 2026 regular_

### Official Review · Reviewer_XWjh · 2026-02-17

**Soundness:** 3
**Presentation:** 3
**Significance:** 3
**Originality:** 1
**Overall Recommendation:** 5
**Confidence:** 3

**Summary:**

The paper proposes **TRAP**: a novel adversarial attack framework for reasoning Vision-Language-Action models (VLAs) that employ Chain-of-Though (CoT) reasoning for predicting actions. TRAP is the first targeted adversarial attacks against VLAs.

The authors find that much of the decision making in VLAs happens in during reasoning, so they use the CoT as the main attack vector. The method optimizes for a patch that will then be printed and put in the physical environment. The patch is optimized such that, when placed in the environment at a given position, the model will produce an attacker-defined reasoning trace (CoT) and produce an attacker-defined action. Further measure are taken to ensure "physical robustness" when printing and placing the patch in the real world: Homography transformation to properly project the patch in the environment, Color smoothing to reduce color distortion due to printing, expectation under transformation to mitigate effects of color distortion after printing.

**TRAP** is tested on a variety of simulated environments as well as in a physical environment, where it is found to yield higher average Attack Success Rate (ASR) compared to baselines which only focus the attack on the CoT [1] or on the Action [2]. Finally, the paper features a large number of ablation studies: the optimization for a universal patch, varying the patch size, generalization to unseen layouts, paraphrasing of instructions).


**References**

[1] Qi, X., Huang, K., Panda, A., Henderson, P., Wang, M., & Mittal, P. (2023). Visual Adversarial Examples Jailbreak Aligned Large Language Models. https://arxiv.org/abs/2306.13213

[2] Wang, T., Han, C., Liang, J. C., Yang, W., Liu, D., Zhang, L. X., Wang, Q., Luo, J., & Tang, R. (2025). Exploring the Adversarial Vulnerabilities of Vision-Language-Action Models in Robotics. https://arxiv.org/abs/2411.13587

**Compliance With Llm Reviewing Policy:**

Affirmed.

**Final Justification:**

The authors clarified how their work is differs from previous literature, and also provided additional experiments that greatly enhance their study. They agreed to reframe their contribution to "the first targeted behavior-hijacking attack against VLAs" and to better clarify the threat model details, as well as fixing minor typographic errors and releasing their source code.

Given the quality of the rebuttal, I am increasing my score to "Accept".

**Key Questions For Authors:**

- Upon examining [1] more closely, do you still claim your work to be the first to propose targeted attacks against VLAs?
- In your understanding, is it possible that the attacks would be less successful in the presence of scissors (or other possibly dangerous items) in simulated environments? It would be great to see an experiment with a knife in the simulated environment too, but I do understand that this may require more time than the short rebuttal period.
- In lines 150-151, you claim "We first collect original tuples (I,R) online...". Can you please provide more informations (e.g. link in footnote, citation to paper, ...) about such data?
- There are important drops in ASR for InstructVLA under paraphrasing compared to MolmoACT. Do you have any intuition as to why this may be?





[1] Wang, T., Han, C., Liang, J., Yang, W., Liu, D., Zhang, L. X., Wang, Q., Luo, J., & Tang, R. (2025). Exploring the Adversarial Vulnerabilities of Vision-Language-Action Models in Robotics. Proceedings of the IEEE/CVF International Conference on Computer Vision (ICCV), 6948–6958.

**Limitations:**

I would state in the paper the additional limitations that the attacker, in my understanding, has:
 - estimation of the Homography transformation (access to camera frames)
 - access to a print-estimation model (or train it, which requires a dataset)

**Strengths And Weaknesses:**

# Strengths
**TRAP** represents a novel attack on VLAs, which constitutes the main novelty of this work. Furthermore, the method was shown to be effective in a real world scenario, which increases the relevance of this work and urges researchers to secure reasoning VLA systems that manipulate the physical world. The paper is also technically sound and provides a vast amount of ablation studies (simulated environments, real environment, patch size, paraphrased instructions, analysis on transferability, universal patch).

# Weaknesses
The main weakness of this work is the limited novelty compared to related works [1,2]. In particular, the authors claim "To the best of our knowledge, this is the first work on the targeted adversarial attack against VLAs" (lines 085-087). However, [1] also features a targeted version of their attack (section 3.4), and the authors do not address how their method compares to the one in [1]. In my understanding, the main difference is the presence of the "CoT Hijacking Loss" term, which is proven to be beneficial. However, if this is the case, I urge the authors to refrain from claiming that their work represents the first application of a targeted attack on VLAs.

Another weakness of this work is the lack of potentially harmful objects in most of the experiments. While in the real world the authors successfully attack a VLA to pick up a (toy) knife instead of a (toy) carrot in 5/15 experiments, in my understanding, most of the results are obtained in simulated environments where knives or other potentially harmful objects (e.g. knife, scissors, hot liquid, ...) are not present. Intuitively, aligned VLMs and VLAs may be more robust to adversarial attacks that have as goal the manipulation of a knife with respect to e.g. an orange.

### Minor weaknesses
 - While experiments in _Table 3_ confirms that CoT is key for the action prediction (mostly for MolmoAct and GraspVLA), I believe that the preliminary analysis in _Section 4_ is made irrelevant by the fact that both instruction masking and cross-sample shuffling effectively represent out of distribution (OOD) data for the VLAs, which are trained on coherent instruction and reasoning data pairs. The results in Table 1 are thus somewhat inadequate to make claims on whether instruction or CoT are more important for making action predictions.

 - The paper does not list some limitations of the attacker which are left implicit. The attacker, other than having white-box access to model weights and gradients, must estimate the Homography transformation and have access to a print-estimation model (or train it, which requires a dataset). I would state these more explicitly in the paper.

 - Typo in caption for Table 8: "Ouuur" instead of "Our".

- The authors do not provide the source code for their experiments.


# Assesment

 - **Soundness**

The work is technically sound and transparent.

 - **Presentation**

The work is for the most part well presented and easy to follow.


 - **Significance**

The problem addressed in the paper is important, and the finding that reasoning VLAs which use CoT expose an additional attack vector is relevant to the scientific community.

 - **Originality**

The paper is only marginally different with respect to previous work [1].


[1] Wang, T., Han, C., Liang, J., Yang, W., Liu, D., Zhang, L. X., Wang, Q., Luo, J., & Tang, R. (2025). Exploring the Adversarial Vulnerabilities of Vision-Language-Action Models in Robotics. Proceedings of the IEEE/CVF International Conference on Computer Vision (ICCV), 6948–6958.

[2] Lu, H., Yu, Y., Yang, Y., Yi, C., Zhang, Q., Shen, B., Kot, A. C., & Jiang, X. (2025). When Robots Obey the Patch: Universal Transferable Patch Attacks on Vision-Language-Action Models. https://arxiv.org/abs/2511.21192

---

> ### Author Rebuttal · Authors · 2026-03-31
>
> ## W1 & Q1. Novelty relative to prior VLA attack work
>
> We thank the reviewer for highlighting this related work. We agree with your assessment and will update our manuscript to accurately frame our contribution as "the first targeted behavior-hijacking attack against reasoning VLAs."
>
> Importantly, our work fundamentally diverges from the cited literature in both **behavioral granularity** and **attack objective**. While previous studies primarily aim to degrade task performance by optimizing specific action dimensions toward fixed values (such as zeroing them out), our objective is far more complex. We focus on the fine-grained, multi-step manipulation required to actively hijack reasoning VLAs. Consequently, our attack induces specific, attacker-defined semantic behaviors by targeting the full spectrum of action dimensions as well as the CoT term, rather than just a few.
>
> ## W2 & Q2. Harmful objects
>
> We appreciate the reviewer's insightful suggestion. Since existing VLAs generally lack safety alignment mechanisms, we initially utilized standard object assets to establish the baseline vulnerability of these models.
>
> However, we agree that demonstrating attacks with a broader range of objects enhances the practical significance of our threat model. Therefore, we have expanded our evaluation to include hazard-oriented scenarios. We conducted additional real-world experiments and introduced dangerous items, such as knives, into our simulation tasks. Empirical observations confirm that our attack can effectively manipulate VLAs into interacting with hazardous objects. We will include these comprehensive results and further discussions in the revised version of the paper.
>
> ## W3. Preliminary study
>
> We frame our masking and shuffling as active interventions to determine if manipulating intermediate features, such as CoT, materially affects the final action. This method closely resembles activation patching used in LLM mechanistic interpretability. Therefore, our preliminary study aligns well with our threat model, as the attack fundamentally relies on creating an instruction-CoT mismatch.
>
> ## W4. Threat model details
>
> We thank the reviewer and agree that executing a reliable physical attack requires specific prior knowledge of the deployment setup, including homography estimation and a print-calibration model. We will explicitly detail these attacker-side assumptions in the revised threat-model section.
>
> At the same time, we note that obtaining such priors does not fundamentally expand the attacker beyond the physical-patch threat model we already assume: for example, homography estimation can be obtained through simple calibration-board placement and camera capture in the deployment environment.
>
> ## W5. Typography error
>
> Thank you for catching the typo. We will fix it in the revision.
>
> ## W6. Code avaliability
>
> We are currently organizing our experimental code and plan to open-source it alongside the revised manuscript.
>
> ## Q3. Clarification of tuples
>
> We apologize for the ambiguity regarding the term 'online.' Here,  'online' refers to the interactive data collection process where the VLA model performs rollouts within the environment. This allows us to collect task instructions and their corresponding CoT reasoning.
>
> ## Q4. InstructVLA drop more under paraphrasing
>
> We hypothesize InstructVLA is more sensitive to paraphrasing because its CoT relies on text-like subtask decomposition. In contrast, MolmoACT is more robust to linguistic variation because its internal reasoning relies on trajectory-related tokens.
>
> ---
>
> **Videos of the above physical experiments are available on [this anonymized webpage](https://anonymous.4open.science/w/web-FE44/).**

---

> > ### Author Rebuttal · Reviewer_XWjh · 2026-04-01
> >
> > I thank the authors for their rebuttal, answering my questions and addressing the weaknesses I raised. The authors clarified how their work is differs from previous literature, and also provided additional experiments that greatly enhance their study. They agreed to reframe their contribution to "the first targeted behavior-hijacking attack against VLAs" and to better clarify the threat model details, as well as fixing minor typographic errors and releasing their source code. I apologize for my misunderstanding of the term "online", which meaning is now obvious in hindsight.
> >
> > While I still consider the paper to be only marginally novel and believe that the experiments could be enhanced in a future work (detectability of patches in real-world setting, longer horizons with novel VLAs, ...), the contributions made by the authors are overall interesting, valuable, and offer insight into what will probably become a very highly active area of research in the future. Given the satisfactory rebuttal, I am willing to increase my score to "Accept".

---

> > > ### Author Response · Authors · 2026-04-04
> > >
> > > We sincerely thank the reviewer for the thoughtful follow-up comments and for increasing the score to **Accept**. We greatly appreciate your time and effort in reconsidering our paper, and we are glad that our rebuttal helped clarify the distinctions between our work and prior literature.
> > >
> > > We are especially encouraged that the reviewer now recognizes our main contribution more clearly: this work is the **first targeted behavior-hijacking attack against CoT-based VLA systems**. More broadly, our findings show that CoT in reasoning VLAs can serve not only as a capability-enhancing mechanism, but also as a new attack surface that can be manipulated to induce specific adversary-chosen behaviors. We also hope that our work can help raise broader awareness of security risks in embodied AI and motivate further research on building safer embodied AI systems.
> > >
> > > We also appreciate the reviewer’s constructive suggestions on future directions, such as improving evaluation on patch detectability and longer-horizon settings. We will incorporate these points, along with the clarified threat model, improved framing of contributions, minor corrections, and code release, in the final version.
> > >
> > > Once again, we sincerely thank the reviewer for the valuable feedback, the positive reassessment, and the encouraging support for our work.

---

### Official Review · Reviewer_BWXr · 2026-03-10

**Soundness:** 2
**Presentation:** 2
**Significance:** 2
**Originality:** 2
**Overall Recommendation:** 3
**Confidence:** 4

**Summary:**

This paper reveals a novel security vulnerability within the Chain-of-Thought (CoT) reasoning mechanism of Vision-Language-Action (VLA) models. The authors first demonstrate through empirical analysis that the intermediate CoT reasoning often exerts stronger control over the action generation than the initial user instruction. Based on this finding, the authors propose TRAP (CoT-Reasoning Adversarial Patch), the first targeted adversarial attack framework for VLA models. By placing adversarial patches in the physical environment, TRAP can "hijack" the VLA's internal CoT reasoning, thereby manipulating the robot to execute malicious physical actions (e.g., altering a "hand the apple" instruction to "hand the knife") without modifying the user's text prompt. The paper conducts extensive simulation experiments across three representative VLA models covering both discrete and continuous action spaces, and validates the feasibility and effectiveness of physical attacks on a real Franka robotic arm.

**Compliance With Llm Reviewing Policy:**

Affirmed.

**Final Justification:**

Thanks for the authors with the further clarification regarding patch stealthiness and defense latency. Compared with the previous version of the patch, the current design is indeed more stealthy, and my concern about defense latency has been addressed. However, as the authors themselves acknowledge, the stealthiness of the proposed patch is only comparable to that of the VLA attack baseline and does not demonstrate a sufficient advantage. I therefore still view this as a weakness of the paper. In addition, the newly introduced optimization strategies, such as PGD+TV, constitute a fairly substantial change from the original submission, which in my view would require significant revision of the main paper and may be difficult to properly incorporate within the current review cycle. After carefully weighing these points and considering the opinions of the other reviewers, I have decided to revise my final recommendation to Weak Reject.

**Key Questions For Authors:**

1.The experiments heavily rely on clean, uncluttered tabletop environments. How does TRAP perform in typical, messy real-world scenarios with significant background clutter and occlusions?

2.Figure 1 appears rudimentary and cartoonish. Can the authors provide a more professional illustration that better represents the technical pipeline for the final version?

3.Regarding physical robustness, the current EoT optimization primarily considers static homography transformations and color deviations. If the robot were mobile (e.g., dynamic camera movements), how would the stability of the TRAP attack be affected?

**Limitations:**

yes. The authors have adequately acknowledged the limitations regarding short-horizon tasks and the lack of stealthiness in Section 7 (Discussion), meeting standard academic expectations.

**Strengths And Weaknesses:**

Strengths

Novel Attack Vector: While existing attacks on VLAs are mostly untargeted or confined to the basic perception layer, this paper astutely identifies the CoT reasoning mechanism as an attack medium. Proposing a targeted attack broadens the research boundary for adversarial attacks on embodied AI models.

Rigorous Empirical Foundation: Before proposing the attack framework, the authors conduct preliminary causal analyses in Section 4 (Instruction Masking and Cross-Sample Shuffling). This confirms the dominant role of CoT in VLA decision-making, providing a solid foundation for the subsequent CoT-targeted attack design.

Real-world Physical Deployment: The authors evaluate their method not only on three representative VLA models in simulation but also successfully deploy the physical attack on a real Franka robotic arm. This provides strong evidence that such security threats genuinely exist in the real world.

Weaknesses

Oversimplified Evaluation Scenarios：The currently evaluated tasks are primarily short-horizon, daily pick-and-place operations, which typically do not pose severe threats to humans. The paper fails to demonstrate the effectiveness of TRAP on more complex, multi-step reasoning tasks.

Lack of Stealth in Physical Patches: The adversarial patches used in the current experiments are visually conspicuous. In realistic scenarios, such abnormal objects would easily be noticed and removed by humans, lowering the credibility and practical threat of the attack in the wild.

Incomplete Narrative Lacking Defense Validation: Although the primary focus is on exposing a vulnerability, the defense mechanism proposed in Section 7 remains entirely theoretical. The completeness of this paper would be significantly enhanced if a simple baseline defense experiment were included to validate the feasibility of their proposed countermeasure.

---

> ### Author Rebuttal · Authors · 2026-03-31
>
> ## W1. Oversimplified Evaluation Scenarios
> We agree that it is ideal to evaluate on long-horizon tasks, but current open-source reasoning-VLAs are functionally bound to short-horizon pick-and-place manipulation. During rebuttal, we tested MolmoACT and InstructVLA on Simpler-Instruct long-horizon tasks. Even under benign settings, both failed after executing only the first pick-and-place subtask. Consequently, TRAP cannot induce long-horizon malicious behavior if the victim model itself cannot reliably perform the benign equivalent.
>
> However, short task horizons do not equate to low practical risk. Our real-world experiments demonstrate that hijacking a simple primitive can force a robot to execute a safety-critical failure, such as grasping a dangerous object (e.g., a knife instead of a carrot). We will clarify this critical distinction between task horizon and practical threat severity in the revision.
> ## W2. Lack of Stealth in Physical Patches
> We fully agree that stealth is vital for realistic physical deployment. To address this, we conducted new experiments during the rebuttal, improving the physical design in two ways:
> - Semantic Disguise: We disguised the patch as plausible everyday items (e.g., desk mat, mouse pad, or a coaster).
> - Texture Regularization: We applied semantic constraints derived from VGG and CLIP during optimization, ensuring the patch exhibits natural textures rather than conspicuous artificial patterns.
>
> Both simulation and physical tests confirm these disguised patches maintain attack efficacy while drastically improving visual plausibility. We will include these updated examples in the revision.
> ## W3. Missing Defense Validation
> Thank you for your suggestion. Adding a baseline defense significantly improves the paper. As theoretically proposed in Section 7, we implemented a lightweight defense to verify alignment between the task intention latent in the CoT and the initial user input. During the rebuttal, we utilized external Vision-Language Models (VLMs) as independent detectors. Evaluated on a binary dataset of benign and attacked trajectories, we measured the Benign Consistent Rate (True Negatives) and Attack Reject Rate (True Positives):
>
> | Defense VLM | Victim VLA | Benign Consistent Rate | Attack Reject Rate |
> | :--- | :--- | ---: | ---: |
> | Qwen2.5-VL-3B | GraspVLA | 23.0% | 85.5% |
> | Qwen2.5-VL-3B | MolmoACT | 24.1% | 99.4% |
> | Qwen2.5-VL-3B | InstructVLA | 65.6% | 100.0% |
> | GPT-4o | GraspVLA | 78.0% | 72.0% |
> | GPT-4o | MolmoACT | 89.7% | 66.0% |
> | GPT-4o | InstructVLA | 92.2% | 87.1% |
> | GPT-5 | GraspVLA | 75.0% | 93.8% |
> | GPT-5 | MolmoACT | 82.8% | 79.4% |
> | GPT-5 | InstructVLA | 95.5% | 100.0% |
>
> The preliminary results show that while smaller models (Qwen) struggle with high false-positive rates (indicated by low benign consistency), a strong detector like GPT-5 identifies hijacked CoTs with high accuracy (up to 100% rejection on InstructVLA) while maintaining robust benign consistency. We will add this experiment to validate the feasibility of our proposed countermeasure.
> ## Q1. Clutter and Occlusion Robustness
> We fully agree that occlusion is a critical real-world factor, which we evaluated from two perspectives:
> - Standard Patches: Synthetic masking reveals that while small corner occlusions allow partial attack success, large center occlusions degrade it. This aligns with broader adversarial-patch literature.
> - Disguised Patches in Clutter: Leveraging the stealth design from W2, we evaluated a large "tablecloth-style" patch under highly cluttered, multi-object tabletop arrangements. This format remained robust to changing object layouts, unseen distractors, and partial patch coverage.
>
> These results suggest that while naive patches are sensitive to central occlusion, realistic large-surface patches function reliably under clutter. We will add these findings to the appendix.
> ## Q2. Figure 1 Quality
> Thanks for your valuable suggestion. We have redrawn Figure 1 into a professional technical pipeline diagram that better conveys the CoT-hijacking architecture. We will replace the cartoon-like version in the final version.
> ## Q3. Dynamic Camera Motion
> Thanks for your thoughtful comment. Our current Expectation over Transformation (EoT) covers static planar homography and color shifts, matching the fixed third-person camera setups of the evaluated VLAs. If deployed on highly dynamic mobile robots, the EoT process must be extended to simulate continuous viewpoint changes and motion-induced blur. We will clarify our static-camera assumption and discuss dynamic-view robustness as future work.
>
> ---
>
> **Videos of the above physical experiments are available on [this anonymized webpage](https://anonymous.4open.science/w/web-FE44/).**

---

> > ### Author Rebuttal · Reviewer_BWXr · 2026-04-01
> >
> > I acknowledge the authors' extensive efforts during the rebuttal phase and appreciate the professional redrawing of Figure 1, which has meaningfully improved the paper's presentation quality. However, after carefully reviewing the visual evidence provided on the anonymized webpage and evaluating the proposed defense, I found that core flaws regarding physical stealthiness and real-world applicability remain fundamentally unresolved. Specifically regarding stealthiness (W2 & Q1), while the rebuttal claims the patches were updated with "semantic disguise" and "natural textures," the actual gallery on the website displays patches—such as the large tablecloth and floral patterns—that are still saturated with conspicuous, brightly colored high-frequency adversarial noise. These objects look highly unnatural in a physical environment and would immediately raise suspicion, indicating that the paper severely overclaims the feasibility of this attack in realistic deployments. Furthermore, while the baseline defense (W3) is a useful conceptual addition, relying on massive, cloud-based VLMs like GPT-4o or GPT-5 for real-time consistency detection is practically unviable for embodied robotic control, which requires extremely low latency and relies heavily on on-board edge computing. Because the visual evidence directly contradicts the textual claims of stealth and the defense assumes unrealistic deployment conditions, these fundamental issues cannot be thoroughly resolved in a short rebuttal phase. Therefore, I maintain my original score.

---

> > > ### Author Response · Authors · 2026-04-04
> > >
> > > We sincerely thank the reviewer for the insightful comments. We are encouraged that the revised Figure 1 and the supplementary experiments helped address part of your concerns. Below, we further clarify the two main issues you raised regarding **physical stealthiness** and the **practicality of the defense in real-world deployment**.
> > >
> > > ### Q1: Physical Stealthiness
> > >
> > > We acknowledge and have improved the **physical stealthiness** as well as the **naturalness** of the physical adversarial patches.
> > >
> > > (1) **High-frequency adversarial artifacts**: To improve stealthiness, we first reduce the **proportion of high-frequency adversarial artifacts**,  by increasing the weight of the **TV** regularization during patch optimization (TV loss weight = 100). We incorporated **Deep Image Prior (DIP)** and **PGD** respectively.  Empirically, the **PGD+TV** method substantially reduced the high-frequency components in the generated patches, and the measured high-frequency ratio are decreased by **98.66%**, i.e., 0.2381 $\rightarrow$ 0.0038 compared to the original result (PGD with  TV loss weight = 0.5), as shown below:
> > >
> > > |                      | Orignal | DIP + TV | PGD + TV |
> > > | -------------------- | ------- | -------- | -------- |
> > > | High-frequency ratio | 0.2831  | 0.0250   | 0.0038   |
> > >
> > > (2) **Reference-image-aided optimization**: To further improve naturalness, we constrained the optimization with reference images corresponding to realistic semantic patterns (e.g., a sports car or Iron Man). In paticular, we increased the weight of **VGG-based perceptual loss** using reference images. The results show that our method can still achieve effective attacks while preserving the naturalness and semantics of the patch (see Fig.2 and Fig3. on the website).
> > >
> > > Combining the two methods above, we compared our improved patches with prior VLA adversarial patch attacks published at top venues [R1] [R2]. These comparisons suggest that our optimized patches are on par with existing alternatives in terms of visual naturalness. We have updated the webpage with the new patch visualizations, real-world attack videos, and side-by-side comparisons with prior work.
> > >
> > > [R1]: Wang T, Han C, Liang J, et al. "Exploring the adversarial vulnerabilities of vision-language-action models in robotics", ICCV, 2025.
> > >
> > > [R2]:  Lu H, Yu Y, Yang Y, et al. "When Robots Obey the Patch: Universal Transferable Patch Attacks on Vision-Language-Action Models.", CVPR, 2026.
> > >
> > > ### Q2: Latency and Practicality of the Defense
> > >
> > > We appreciate the reviewer’s concern regarding the latency and deployability of the defense. While the primary goal of our work is to reveal the vulnerability of reasoning VLAs, we agree that practical defenses should be compatible with **low-latency embodied control**.
> > >
> > > (1) **Lightweight defense tailored to different reasoning formats**: Our original defense was designed with **generality** in mind, since reasoning VLAs produce heterogeneous chains of thought, including bounding boxes, traces, and free-form text. To address the deployment concern more directly, we further designed **lightweight consistency checks tailored to different reasoning formats**: 1) For **bouding-box-based COT** (e.g.,  GraspVLA ), we use an open-vocabulary detector to verify consistency between the instruction and the predicted box, 2) For **trace-baed COT** (e.g, MolmoACT), we check the consistency between the predicted trace and the detected target object, 3) For **textual-based COT** (e.g., InstructVLA), we  use a lightweight text encoder to measure the semantic similarity between the instruction and the generated reasoning. We have validate the performance of the proposed lightweight defenses, and the resulting performance is summarized below:
> > >
> > > | CoT Category          | BCR (vs. GPT-5) | ARR(vs. GPT-5)  | Time Consumption |
> > > | --------------------- | --------------- | --------------- | ---------------- |
> > > | Bouding-box-based COT | 87.00% (+12%)   | 92.50% (-1.8%)  | 201.67 ms        |
> > > | Trace-baed COT        | 77.01% (-5.79%) | 87.42% (+8.02%) | 114.35 ms        |
> > > | Textual-based COT     | 86.07% (-9.43%) | 91.94% (-8.60%) | 2.62 ms          |
> > >
> > > BCR=Benign Consistent Rate , ARR=Attack Reject Rate,
> > >
> > > (2) **Defense results**:  These results show that the proposed lightweight defenses achieve **performance close to GPT-5-based checking while incurring only millisecond-level overhead**. Compared with the runtime of the reasoning VLA models, the overhead of our lightweight defenses is small; in the most efficient case, it is only **0.20%** of the model inference time (for Textual-based COT, with ~1.3s inference time).
> > >
> > > We thank the reviewer again for the thoughtful feedback. We hope that the additional experiments and clarifications help address your concerns regarding both stealthiness and defense practicality.
> > >
> > > ---
> > >
> > >  **Videos of the above physical experiments are available on [this anonymized webpage](https://anonymous.4open.science/w/web-FE44/).**

---

### Official Review · Reviewer_5UjY · 2026-03-15

**Soundness:** 3
**Presentation:** 3
**Significance:** 3
**Originality:** 3
**Overall Recommendation:** 3
**Confidence:** 4

**Summary:**

This paper introduces TRAP, a physical adversarial patch targeting Chain-of-Thought (CoT)-enabled VLA models to hijack robot behavior without altering user instructions. The core premise is that jointly attacking the intermediate CoT reasoning and downstream actions is more effective than targeting actions alone. Evaluated across three VLA architectures and a small real-world setup, the work highlights a timely and non-trivial security vulnerability. However, while the problem is important, the empirical evidence and practical threat model currently lack the breadth and strength needed to fully support the paper's broader claims.

**Compliance With Llm Reviewing Policy:**

Affirmed.

**Key Questions For Authors:**

1. Can authors provide stronger black-box evidence beyond fine-tuned-variant transfer?
If the attack remains effective across unseen architectures or stronger surrogate-transfer settings, it would materially strengthen the practical significance of the paper.

2. How effective is TRAP on genuinely long-horizon reasoning tasks?
The paper currently admits it has only been tested on pick-and-place primitives; if the method also works on longer-horizon settings, that would substantially improve my assessment of the paper’s scope.

3. Can authors redo the instruction-variant experiment using only truly semantically equivalent paraphrases?
This would clarify whether the current robustness claim reflects real linguistic transfer or is partly confounded by instruction changes that alter task meaning.

**Limitations:**

See weakness and key questions

**Strengths And Weaknesses:**

## Strengths

1. The paper asks an important security question.
The shift from generic failure attacks to targeted control hijacking in CoT-enabled VLAs is meaningful, and the paper correctly points out that explicit intermediate reasoning can expose a new manipulation surface.

2. The attack formulation is clear and technically coherent.
The paper combines a CoT hijacking loss with an action loss, and explicitly adapts the action objective to both discrete-token and continuous-action VLAs. It also includes physically motivated components such as homography-based placement, smoothing, calibration, and EoT.

3. The evaluation includes both simulation and real setup.
The paper tests three reasoning VLA families with different CoT forms and includes a physical patch demonstration, which is valuable for a security paper.

## Weakness
1. The empirical scope is limited relative to the paper’s claims.
The main evaluation uses only three victim models and five manipulation tasks, with 25 layouts for training and 10 for testing; the real-world evidence is a single hazardous-redirection scenario with 15 trials. For a paper making broad claims about reasoning-VLA security, this is still a fairly small and curated empirical base.

2. The claimed generalization is overstated.
Some of the most interesting robustness numbers are actually weak: transfer drops sharply relative to the optimized setting, and the “universal patch” mostly fails under scene-relevant instructions. This makes the attack look much more task- and model-calibrated than the paper’s broader framing implies.

3. The preliminary “competition mechanism” analysis is suggestive, but not fully convincing.
The masking and shuffling interventions are interesting, but the results are quite heterogeneous across models: InstructVLA and MolmoAct look nearly balanced, whereas GraspVLA is heavily CoT-dominated. That is enough to motivate the attack, but not enough to establish a clean, general causal account of how instruction and CoT are arbitrated across reasoning VLAs.

4. Some robustness analyses are not cleanly designed.
The instruction-variant study is hard to interpret because the reported “paraphrasing” set includes prompts such as “place the orange,” which is not a paraphrase of “pick orange.” This weakens the claim that the attack is robust to benign linguistic variation.

5. TRAP does not consistently demonstrate a decisive advantage over simpler CoT-level manipulation.
On MolmoACT, TRAP is essentially tied with the CoT-only baseline in ASR, and on GraspVLA the gap is modest. The method seems clearly most helpful on InstructVLA, where the action loss mainly stabilizes mode collapse. That makes the contribution look more like a model-dependent improvement than a uniformly stronger attack paradigm

---

> ### Author Rebuttal · Authors · 2026-03-31
>
> ## W1. Experimental Scope
> We appreciate the valuable feedback. Evaluating reasoning-VLAs requires balancing rigor with the severe computational realities of this emerging field:
> - Models: Reasoning-VLAs are novel with very few open-source options. The three evaluated models represent the currently available SOTA.
> - Tasks & Layouts: Evaluating 5 tasks aligns with core manipulation benchmarks (e.g., SimplerEnv uses 4). Using 25 training and 10 testing layouts (with 5 trials each) deliberately reflects a realistic threat model where an attacker has limited environment access.
> - Compute Bottleneck: Reasoning-VLA inference is rather slow (\~5s/step) and requires many (~100) steps for 1 trial. Our current simulation suite alone demands massive compute overhead: 5 tasks * 35 layouts * 5 trials * 100 steps * 5s = ~5 GPU days per model.
>
> Moreover, we conducted more experiments on more stealthy patches and robustness tests against clutter and occlusion. [Video Link](https://anonymous.4open.science/w/web-FE44/)
> ## W2 & Q1. Generalization & Black-Box Transfer
> We will reframe our transferability results as a preliminary exploration of feasibility rather than strong universality. For Q1, cross-architecture transfer (e.g., MolmoACT to GraspVLA) remains a formidable open challenge due to severe architectural heterogeneity:
> 1. Disparate Action Spaces: Transferring adversarial features between discrete tokenization and continuous diffusion creates a fundamental semantic disconnect.
> 2. Misaligned Modalities: Victim models rely on entirely different visual backbones and CoT formats (textual vs. spatial).
>
> We consider ensemble-based surrogate optimization as a potential solution.
> ## W3. Heterogeneity in the "Competition Mechanism"
> We appreciate your attention to the variance across models. We view this heterogeneity not as a limitation of the analysis, but as a critical characteristic of the current reasoning-VLA landscape. Different models arbitrate the influence of CoT and instructions differently due to three systemic design choices:
> - Reasoning-Action Coupling: GraspVLA tightly couples spatial CoT (bounding boxes) with continuous actions, leading to strict CoT dominance. InstructVLA loosely couples textual subtasks with a downstream policy, resulting in a balanced, competing signal.
> - Training Objectives: Models assign varying loss weights and architectural bottlenecks to reasoning versus action generation during pre-training.
> - Task & Data Distribution: Spatial alignment tasks inherently force a high reliance on CoT, whereas broad semantic tasks maintain a stronger dependency on the original user prompt.
> ## W4 & Q3. Rigorous Instruction-Variant Experiment
> Thank you for pointing out this. We agree that "place the orange" altered task semantics rather than just linguistic variation. As requested (Q3), we redesigned the evaluation using only strictly semantically equivalent instructions (e.g., replacing "place" with exact synonyms like "grab"). The updated results of Table 4 are below:
>
> | Setting | MolmoACT ASR (%)| MolmoACT Score | InstructVLA ASR (%)| InstructVLA Score |
> | --- | --- | --- | --- | --- |
> | Original | 72.0 | 0.34 | 67.4 | 0.56 |
> | Paraphrasing | 70.6 | 0.32 | 25.1 | 0.09 |
>
> By removing the semantic shifts, the attack performance becomes even better. Notably, on MolmoACT, TRAP demonstrates strong robustness to paraphrasing, maintaining a 70.6% ASR (almost no degradation from 72.0%). We will update Table 4 and its corresponding discussion with this corrected setup.
> ## W5. The Role of CoT-Only vs. TRAP
> Thank you for your thoughtful comment. "CoT-Only" is not a prior baseline; it is our own ablation representing the core vulnerability we discovered. Its high effectiveness on MolmoACT and GraspVLA proves our central thesis: hijacking reasoning manipulates physical actions.
>
> We introduced TRAP (joint CoT+Action loss) specifically to address the heterogeneity of reasoning-VLAs (W3). In models like InstructVLA, attacking CoT alone conflicts with benign instructions, causing severe mode collapse (Sec 6.2). TRAP resolves this by constraining the action manifold. Thus, TRAP isn't an incremental boost, but a generalized formulation ensuring stable attacks across diverse VLA paradigms.
> ## Q2. Effectiveness on Long-Horizon Tasks
> We agree this is a critical frontier. However, current open-source reasoning VLAs inherently struggle with true long-horizon execution.
>
> 1. Model Limits: During rebuttal, we tested MolmoACT and InstructVLA on Simpler-Instruct long-horizon tasks. Even under benign settings, both failed after executing only the first pick-and-place subtask.
> 2. Attack Efficacy: Crucially, TRAP successfully hijacked their intermediate reasoning (subtask planning) in these setups. Since long-horizon execution relies on correct sequential reasoning, subverting this guarantees cascading failure.
>
> We will add these new results to clearly distinguish between current VLA capability boundaries and our attack's scope.

---

> > ### Author Rebuttal · Reviewer_5UjY · 2026-04-04
> >
> > My concerns have been adequately addressed.

---

> > > ### Author Response · Authors · 2026-04-06
> > >
> > > Thank you very much for your thoughtful follow-up and constructive suggestions. We sincerely appreciate the time and effort you invested in reviewing our work, and we are glad that our responses have adequately addressed your concerns. If you find the clarifications and additional evidence helpful, we would be very grateful if you could kindly reconsider your overall score in light of them.

---

### Decision · Program_Chairs · 2026-04-30

**Decision:**

Accept (regular)

**Comment:**

In this paper, the authors propose TRAP, a physical adversarial patch attack that targets Chain-of-Thought (CoT) reasoning in vision-language-action (VLA) models. The core observation is that reasoning VLAs ground their actions heavily in intermediate CoT outputs, so a patch optimized to hijack the CoT can induce attacker-chosen behaviors without modifying the user's text instruction. The method combines a CoT-hijacking loss with an action loss (adapted to both discrete-token and continuous-action VLAs), with physical-robustness components including homography, color smoothing, print calibration, and Expectation over Transformation (EoT). Evaluation covers three reasoning VLAs (MolmoACT, InstructVLA, GraspVLA) on five manipulation tasks in simulation, plus a physical Franka demonstration including a hazardous-redirection scenario (carrot -> knife).

Post-rebuttal reviews split into two weak rejects and one accept (raised from a lower score). Initial concerns covered narrow empirical scope (3 models, 5 tasks, 15 real-world trials on one scenario); overstated generalization, with the universal patch failing on scene-relevant instructions; TRAP not consistently beating the CoT-only baseline on MolmoACT (R-5UjY); limited patch stealth and missing defense validation (R-BWXr); and a preliminary masking/shuffling study that may produce OOD inputs rather than clean causal evidence (R-XWjh). R-XWjh also flagged that the original "first targeted attack on VLAs" framing was contradicted by prior work.

The authors presented a substantial rebuttal: a strict-paraphrase re-evaluation where MolmoACT held at 70.6% ASR versus 72.0% original; improved stealth via PGD+TV with reference-image-guided optimization (~ 99% reduction in high-frequency ratio); a lightweight format-specific defense (~ 90% attack rejection with millisecond-level overhead); hazardous-object scenarios; and a reframed contribution as "the first targeted behavior-hijacking attack against reasoning VLAs." R-XWjh raised to Accept; R-5UjY marked concerns fully resolved but maintained weak reject; R-BWXr maintained weak reject citing remaining stealth concerns, partly answered by the lightweight defenses added in the final-round exchange.

Overall, targeted CoT-hijacking on reasoning VLAs is a new attack vector within the reframed scope, and the physical Franka demo supports practical relevance. The empirical scope remains narrow, the universal patch is limited, and the TRAP-vs-CoT-only gap is model-dependent: but these are scoping concerns rather than fundamental ones, and R-XWjh's raise to Accept reflects the cleaner reframed contribution. I recommend acceptance, and encourage the authors to integrate the PGD+TV stealth improvements, lightweight defenses, corrected paraphrase evaluation, hazardous-object scenarios, and the reframed contribution into the main text, along with an explicit discussion of when TRAP adds value over the CoT-only baseline.